# Plant-Based Fermented Beverages: Development and Characterization

**DOI:** 10.3390/foods12224128

**Published:** 2023-11-15

**Authors:** Marcela Aparecida Deziderio, Handray Fernandes de Souza, Eliana Setsuko Kamimura, Rodrigo Rodrigues Petrus

**Affiliations:** Faculdade de Zootecnia e Engenharia de Alimentos, Universidade de Sao Paulo, Pirassununga 13635-900, SP, Brazil; mdeziderio@usp.br (M.A.D.); handrayfds@usp.br (H.F.d.S.); elianask@usp.br (E.S.K.)

**Keywords:** acid lactic bacteria, fermentation, non-dairy, stability

## Abstract

The production of plant-based fermented beverages has been currently focused on providing a functional alternative to vegan and/or vegetarian consumers. This study primarily targeted the development and characterization of fermented beverages made up of hydrosoluble extracts of oats, almonds, soybeans, Brazil nuts, and rice. The fermentation was carried out by lactic cultures of *Bifidobacterium* BB-12, *Lactobacillus acidophilus* LA-5, and *Streptococcus thermophilus*. Plant extracts were fermented at 37 °C for 12 h, with and without sucrose supplementation. The physicochemical and microbiological stability of the extracts was monitored for 28 days at 5 ± 1 °C. The composition of the fermented beverages was subsequently determined. The pH values measured at the beginning and the end of the extracts’ fermentation ranged between 6.45 and 7.09, and 4.10 to 4.97, respectively. Acidity indices, expressed as a percentage of lactic acid, ranged from 0.01 to 0.06 g/100 mL at the beginning of the fermentation, and from 0.02 to 0.33 g/100 mL upon fermentation being concluded. Most fermented extracts achieved viable lactic acid bacteria counts exceeding 10^6^ CFU/mL during storage. Sucrose supplementation did not alter the rate of bacterial growth. The findings showed that the complete replacement of dairy ingredients with water-soluble plant extracts is a potential alternative for developing a functional fermented plant-based beverage.

## 1. Introduction

The population’s awareness of healthy diets and eating habits has resulted in a growing demand for novel food products. Adherence to this lifestyle depends, in particular, on a balanced diet and the regular practice of physical exercise that promotes a high quality of life, prevention of degenerative diseases such as diabetes and cancer, and increased life expectancy [1,2,3]. 

The food industry has invested in the development of nutritious and functional foods. Functional foods are conventional food products that provide additional health benefits in addition to their nutritional properties whenever they are administered or consumed as part of a standard diet [4]. The demand for these foods has grown consistently with increased health consciousness and life expectancy [3].

Food fermented by lactic acid bacteria (LAB) with probiotic functionality has continuously been an alternative in developing functional foods. Fermentation of food matrices using LAB can improve the product´s functionality and enhance its nutritional quality [5,6,7]. Additionally, a daily intake between 10^6^ to 10^8^ CFU/g of viable probiotic microorganisms is defined as the specific amount providing health benefits to the host, a result of its development within the gastrointestinal tract [8].

The main probiotics genera used in food matrices are *Lactobacillus* and *Bifidobacterium* [9], and dairy products are the most widely available probiotics on the market. Fermented dairy products such as yogurt, fermented milk, and fermented whey beverages constitute the majority of foods containing probiotic cultures [10]; however, the possibility of using plant-based products as probiotic carriers has also been investigated [11].

Cereals, fruits, nuts, grains, and legumes are healthier options for developing novel functional products. These foods exhibit advantages over dairy products, including the absence of lactose, potentially allergenic proteins, and fats. Additionally, they meet the market demand for vegetarian alternatives, dietary restrictions resulting from allergies to proteins contained in cow’s milk, and specific food ideologies [1,12,13]. 

The preparation of fermented plant-based beverages presents great potential in the development of a new functional product [12]. Fruits, vegetables, and cereals can be used as alternative substrates for fermented dairy-based products [14,15]. Plant-based beverages do not exhibit the same nutritional value as milk, but fermentation can enrich the functional aspects of the finished product, and studies that determine the profile of these foods should be exploited, such as microbial species, cell viability, and survival of microorganisms throughout periods of storage, as well as the acceptability of the product in sensory terms. This study targeted the development and viability of mixed lactic acid bacteria cultures containing *Lactobacillus acidophilus* LA-5, *Bifidobacterium* BB-12, and *Streptococcus thermophilus* intended for use in the preparation of fermented plant-based beverages. 

Very few studies report lactic fermentation entirely made up of plant-based extracts, either with or without added sugar. In this study, fermented beverages were prepared with almonds (*Prunus dulcis*), rice (*Oryza sativa* L.), oats (*Avena sativa* L.), Brazil nuts (*Bertholletia excelsa* H.B.K), soybeans (*Glycine max* L.) extracts and cow’s milk as the control. The samples were fermented for 12 h and stored at 4 °C for 28 days, and the pH, acidity, soluble solids, proximate composition, and viability of microorganisms were determined.

## 2. Materials and Methods

Five sources of raw material were used to prepare the plant extracts, namely oats, rice, almonds, Brazil nuts, and soybeans, which were purchased in specialized natural products stores in the cities of Mogi Guaçu, Aguaí, and São Paulo, Brazil. Refined sugar (União^TM^, Pirassununga, Brazil) was used to adjust and standardize the extracts’ soluble solids content to 4.0 °Brix. Ultra high temperature (UHT) long-life whole milk was used as a control. Each of the extracts was prepared separately.

Ready for use lyophilized lactic acid culture was used (BioRich^TM^), which contains *Streptococcus thermophilus*, *Lactobacillus acidophilus* LA-5^®^ (1 × 10^6^ CFU g^−1^), and *Bifidobacterium* BB-12 (1 × 10^6^ CFU g^−1^) cultures and is produced by the Denmark-based company Chr. Hansen A/S and imported by Chr. Hansen Ind. and Com. Ltda de Valinhos—SP, Brazil. Lactic cultures were purchased from the local market in the city of Mogi Mirim—SP, Brazil.

### 2.1. Preparation of Plant Extracts

During the preparation of almond (Figure 1a) and Brazil nut (Figure 1b) extracts, 100 g of each nut were used. Raw materials were first hydrated by being immersed in a container containing 500 mL of water for 12 h at 25 °C. Once this period had passed, the water was discarded, the nuts were washed and the almonds peeled and re-washed. Subsequently, the nuts and almonds were individually crushed in mineral water (potable) for 5 min at 37 °C using a domestic blender, until a homogeneous mixture was obtained. A proportion of 1 L water was used for every 100 g (*m*/*m*) of almonds and Brazil nuts. Next, the mixtures were filtered using previously sanitized cotton cloth filters. The filtrates were then collected in a glass container and pasteurized.

The oat extract (Figure 1c) was obtained were ground with 1 L of water and 80 g of oats (*w*/*w*) at an ambient temperature of 25 °C for 5 min using a household blender until a homogeneous mixture was obtained. Filtration was performed using cotton cloth, with the filtrate collected in a glass bottle and subsequently pasteurized and stored [16].

First, the soybeans were cooked with 500 mL of water to inactivate the lipoxygenase enzymes in boiling water at approximately 100 °C for 5 min. The grains were then separated using a domestic sieve and washed in running water at room temperature. Once enzymatic inactivation was completed, the grains were crushed in a proportion of 100 g of soybeans to 1 L of water (*m*/*m*) in a domestic blender for 5 min until reaching a homogeneous consistency. The soy extract (Figure 1d) was cooked again for another 10 min, filtered through a cotton cloth, stored in a glass bottle, and pasteurized.

In order to obtain the rice extract (Figure 1e), grains were cooked using a proportion of 100 g rice with 200 mL of water (*m*/*m*) for 20 min at 100 °C. Once the grains were cooked, the excess water was drained and the grains were crushed with water using a household blender at a proportion of 100 g of rice cooked to 1 L of water (*w*/*w*) for 5 min until a homogeneous mixture was obtained. The extract obtained was filtered and stored in the same manner used for the remaining extracts [17]. 

The extracts produced (Figure 1) were pasteurized at 80 °C/5 min in Schott^TM^ glass containers in order to eliminate deteriorating microorganisms and non-sporulating pathogens [18]. After pasteurization was complete, the extracts were stored under refrigeration at a temperature of 5 ± 1 °C.

### 2.2. Fermentation 

The extracts obtained were separated into the following three groups:

Group 1: UHT whole milk (control).

Group 2: plant extracts (rice, almonds, oats, Brazil nuts, and soybeans).

Group 3: plant extracts in which the level of soluble solids was adjustable (rice, almonds, oats, and Brazil nuts)—in this group, sucrose was added to the extracts in order to standardize the soluble solids content in the soybean extract, which was the raw plant material with the highest concentration of sugars.

The lactic cultures were initially weighed in a sufficient quantity providing a concentration of 400 mg/L in accordance with the manufacturer’s recommendation. It was then dissolved in 50 mL of the plant extract. After the lactic cultures were solubilized, the culture medium was filled to 250 mL with plant extract. All extracts used were pre-heated in a water bath at 37 °C and the lyophilized culture was, subsequently, inoculated. After inoculation, fermentation was carried out at 37 ± 2 °C without agitation in a biochemical oxygen demand (BOD) incubator. A 12-h fermentation time for plant extracts was established through means of preliminary tests. During fermentation, samples were collected for the purposes of analyzing the concentration of soluble solids, titratable acidity, and pH levels. Once the fermentation process was completed, samples were collected for further analysis. All fermentation was carried out with three replicates. The fermentates that were obtained were filled into 200 mL high-density polyethylene (HDPE) bottles and stored at 5 ± 2 °C for 28 days in order to assess physicochemical stability and cell viability.

### 2.3. Physicochemical Analysis of Fermented Products

Titratable acidity was determined in accordance with the titration method [19]. Ten mL of the fermented sample was transferred to a 125 mL Erlenmeyer flask, and 10 mL of water and 4 drops of the basic acid indicator, 1% (*w*/*v*) phenolphthalein, were added. The sample was titrated using a standardized 0.1 mol/L sodium hydroxide (NaOH) solution and agitated until a permanent pink coloration was identified for 30 s. Results were expressed as a percentage (%) of lactic acid.

The pH values were measured through means of a potentiometer and the use of a digital pH meter (Digmed). The pH meter was calibrated to buffer solutions of pH 7 and 4.50 mL of each fermented sample was used to measure pH levels [19]. A previously calibrated portable refractometer with direct reading (Instrutemp) was used to determine the concentration of soluble solids (°Brix).

### 2.4. Beverages’ Composition

The Bligh-Dyer method was used to determine total lipids, which consists of cold extraction, a technique that does not require heating, carried out using a mixture of three solvents: chloroform, methanol, and water. Then, the amount of dry extract obtained was determined in accordance with the direct oven drying method [19]. Oven drying is based on the removal of water through means of heating using hot air and the absorption of a very thin layer of food beginning with a 10 g sample in a porcelain capsule or Petri dish at a temperature of 105 °C. The sample was subsequently cooled to room temperature in a desiccator and then weighed using an analytical balance. This procedure was repeated until a consistent mass value was obtained [20].

The dry ashing method was used to determine ash content, which consists of incinerating 10 g of the sample in a porcelain capsule in a muffle furnace at a temperature between 550 °C and 570 °C, thereby promoting the evaporation of water, volatile substances, and oxidation of organic matter [20].

Protein content was determined using Kjeldahl digestion [21]. This method is divided into three main stages: (a) sample digestion, (b) distillation, and (c) titration. Protein content is therefore determined based on nitrogen content, in which nitrogen generally corresponds to 16% of the weight of the protein sample.

In order to determine sugar levels, samples were filtered through a nylon syringe filter (0.22 µm) and diluted using a dilution factor of between 50 and 500 times. A 50 µL looping of the sample was injected and the sugar profile was analyzed through means of ion exchange chromatography with pulsed amperometric detection (HPLC-PAD) using a Dionex^®^ chromatograph (Sunnyvale, CA, USA) equipped with a Carbopac PA-1 column (4 × 250 mm), Carbopac PA-1 guard column (4 × 50 mm), GP50 pump, ED40 electrochemical detector, and PEAKNET software. The mobile phases used were: (a) NaOH (50 mMol) and (b) (500 mMol NaOAc + 50 mMol NaOH). The compounds’ peaks were identified through comparison with the retention times obtained in the injected standards. The standards used were glucose, fructose, and sucrose (Sigma-Aldrich, St. Louis, MO, USA).

### 2.5. Lactic Acid Bacteria Count

Fermentates were evaluated for 28 days while stored under refrigeration. Serial decimal dilutions were carried out in 0.1% sterile peptone water in order to determine the lactic acid bacteria count. Dilutions up to 10^−9^ were necessary during this study. 1 mL aliquots were subsequently plated at depth using Man, Rogosa and Sharp (MRS) agar. Plates were incubated at 37 °C for 48 h and colony counts were obtained using direct plate counting [22]. 

### 2.6. Statistical Analysis

The data obtained in the physicochemical, microbiological, and chemical composition analyses were processed using Analysis of Variance (ANOVA) and Tukey’s mean comparison test at 5% significance (95% confidence). For this purpose, Microsoft Excel and Sisvar 5.6 software were used.

## 3. Results and Discussion

### 3.1. Plant Extract Fermentation 

Table 1 shows the results for pH levels determined during fermentation. The existence of four groups of substrates with pH values presenting differences (*p* ≤ 0.05) in relation to initial conditions were identified. Initially, all substrates presented high pH levels (ranging from 6.45 to 7.09), suggesting a reduced concentration of organic acids in the raw materials used. It was also confirmed that adjusting soluble solid concentration by adding sucrose allowed for very little influence on pH levels since only the almond extract presented a significant difference between one form and another (Almonds and Almonds WA).

Final results for pH levels (Table 1), revealed that the entirety of substrates with an adjustment in soluble solids of 4.0 °Brix presented lower and statistically different pH levels (*p* ≤ 0.05) when compared to the original extract, indicating that sucrose supplementation is an important factor in the acidification of the beverages produced. The influence of the addition of sucrose was significant in rice and oats extracts, in which pH values in extracts without adjustment of soluble solids were the highest (pH 4.92 and 4.97, respectively). With regards to supplemented extracts (adjustment of soluble solids of 4.0 °Brix), pH values were lowest (pH 4.15 and 4.10, respectively) when a general comparison was carried out. In terms of acidity, all plant extracts subjected to fermentation presented an increase in this parameter (Table 1). In general terms, when comparing initial acidity values, it was observed that the initial adjustment of soluble solids offered little interference, with the exception of the Brazil nuts extract. Acidity values for all substrates presented significant differences between one other (*p* ≤ 0.05) after fermentation was complete. Brazil nuts in which solids were adjusted (Brazil nuts WA) presented similarities to adjusted almond fermentate (Almond WA). It was also observed that the influence of the adjustment of soluble solids was significant (*p* ≤ 0.05) in the rice substrate (Rice and Rice WA); however, interference with acidity (Almond and Almond WA) was not observed in the almond substrate.

The control sample (milk) obtained the highest acidity index during the fermentation period, and the final acidity was significantly different from the initial acidity (Table 1). The final acidity for the control sample (0.87 g of lactic acid/100 mL) was found to comply with Brazilian legislation, which recommends values for acidity between 0.6 and 2.0 g of lactic acid/100 mL [23].

As can be observed in Table 1, the extracts that reached the highest level of acidity were fermented soybeans, almonds, and Brazil nuts both with (Brazil nuts WA) and without adjustment of soluble solids, the values for which ranged between 0.27 and 0.33 g of lactic acid/100 mL. Soybean, almond, and Brazil nuts extracts without adjustment and almond extracts with adjustment (Almond WA) presented a significant difference in acidity (*p* ≤ 0.05). However, Brazil nuts extract, both with (Brazil nut WA) and without adjustment, did not present a significant difference. Adjustment of the Brazil nuts extract was similar to the almond extract both with (Brazil nuts WA) and without adjustment. Soybean extract presented a final acidity of 0.27 g of lactic acid/100 mL. Fávaro Trindade et al. [24] reported that “yogurt” developed using soybean extract without supplementation reached values for acidity between 0.24% (4 h of fermentation) and 0.33% (6 h of fermentation).

The lowest acidity values obtained were observed for the oat and rice extracts under both conditions, with values ranging between 0.02 and 0.16 g of lactic acid/100 mL. The rice substrate presents differences between the two conditions, with and without adjustment of soluble solids. The oat extract did not present a difference between the original substrate and the one supplemented with sucrose (Table 1). Ghosh et al. [25] assessed an Indian rice fermentate under continuous fermentation for a four-day period at 37 °C. The acidity values observed were 0.01 of raw material and 0.84 g of lactic acid/100 mL on the fourth day of fermentation. Such values demonstrate that fermented rice extract does not present a high acidity index, even if subjected to a long period of fermentation.

The plant-based fermentates that were studied were found to present acidity values that were lower than those recommended under Brazilian legislation [23], which establishes a range between 0.6 and 2.0 g of lactic acid/100 g for fermented milks. Results for soluble solids content are presented in Table 1. The results suggest that, when compared statistically (*p* ≤ 0.05), 3 different groups can be identified at the start of fermentation (0 h). The control sample showed a decrease in soluble solids throughout fermentation, ranging from 13.0 °Brix (start of 0 h) to 7.0 °Brix (end of 12 h) and, when compared to plant-based extracts presented a significant difference (Table 1). The plant-based extracts that presented the most significant reduction of soluble solids were soybean, almond WA, and Brazil nuts WA.

Almond and Brazil nuts extracts in which the soluble solid content was not adjusted remained constant during fermentation (Table 1). The extracts cited therefore presented a reaction to the addition of sucrose, showing significant differences (*p* ≤ 0.05) and provided evidence of the manner in which supplementation acts upon these substrates. Rice and oat extracts were not affected by the addition of sucrose and were shown to remain constant during fermentation under both conditions.

### 3.2. Physicochemical Stability 

Table 2 points out pH behavior during storage under refrigeration at 5 °C. As can be observed from the table, pH values ranged from between 3.63 and 4.50 (initial) and 3.71 and 5.08 (final). The control sample presented an initial pH and final pH of 4.37 and 4.15, respectively. Unadjusted Brazil nuts extract positioned itself near the control, with initial and final values of 4.36 and 4.43, respectively. When both extracts are compared statistically, their initial values do not present a significant difference; however, at 28 days of storage, the unadjusted Brazil nuts extract differed significantly from the control (Table 2).

Soybean extract had the highest initial and final pH at 4.50 and 4.66, respectively. With regard to the remaining extracts, fermented soybean extract presented a significant difference (Table 2). Similar values were obtained by Battistini et al. [26] when evaluating soybean extract supplemented with inulin and fructooligosaccharides using *L*. *acidophilus*, *Bifidobacterium animalis,* and *S*. *thermophilus* cultures.

The unadjusted almond extract presented the lowest initial pH value (4.05), showing little variation and increasing slightly beginning on the seventh day of storage (Table 2). Adjusted almonds showed little variation, with initial and final values of 4.06 and 4.18, respectively. Bernat et al. [27] developed a non-dairy fermentate with an almond extract base, which was supplemented with inulin, fructose, and glucose and stored for 28 days at 4 °C, obtaining pH values equivalent to 4.65 (initial) and 4.65 (final). These results are lower than those obtained in this study.

The pH values for unadjusted oat extract without adjustment varied throughout the 28-day period, moving from 4.17 (initial) to 4.38 (final). Adjusted oat extract also showed variation in pH levels during storage; however, the pH decreased from 3.82 to 3.71, demonstrating that supplementation led to a significant difference between substrates (Table 2). Gupta et al. [28] developed an oat-based fermentate that was supplemented with sucrose and stored for 21 days at 4 °C. The pH values obtained by these authors were 4.5 (initial), and pH remained above 4.00 for 21 days. These values were similar to those obtained for unadjusted oat extract and are higher than those recorded for adjusted extract during this study.

The pH for unadjusted rice extract presented increased variations during storage (Table 2). An increase in pH was observed during the 28 days for this extract, with values ranging between 4.07 and 5.08. However, the fermented extracts generally presented significant pH variations during storage (Table 2).

Acidity behavior during storage under refrigeration at 5 °C, is shown in Table 3. The extracts’ acidity level generally presented little variation during storage. During the course of the study, it was verified that the acidity of the control sample adheres to the established minimum value for fermented milks, which is between 0.6 g and 2.0 g of lactic acid/100 g [23].

The acidity of fermented soybean presented little variation during storage (Table 3); however, when compared to the remaining extracts, the fermented soybean presented a significant difference throughout the storage period, with the exception of the twenty-first day, in which it was found to be statistically equal to unadjusted fermented Brazil nuts extract. Mohammadi et al. [29] obtained similar values for soybean fermentate, with values ranging from between 0.28 and 0.31% for initial lactic acid and 0.29 to 0.41% for lactic acid at the end of a 21-day storage period at 5 °C.

Oat fermentate did not present a significant variation throughout the 28-day period under both conditions (Table 3). Gautam and Sharma [30] prepared a mixture of fermented cereals that were subject to 24 h of fermentation at 37 °C and subsequently stored at 4 °C. The acidity values observed ranged from between 0.09% and 0.18% (*m*/*m*) and were similar to those obtained during this study. Gupta and Bajaj [31], when studying an oatmeal and honey fermentate stored at 4 °C for 4 weeks, obtained an increase in acidity ranging from between 0.90% and 1.74% *m*/*v* during the 4-week period. The rice extract presented the lowest acidity values under both conditions (Table 3). Additionally, the final values for the rice extract’s acidity under assessed conditions did not present a significant variation. During the assessment of a rice-based fermented beverage popularly known as Chicha, Puerari et al. [32] obtained a value of 0.08 g/mL of lactic acid during 12 h of fermentation by lactic acid bacteria. Jukonyte et al. [33], when evaluating a rice-based beverage, obtained contents ranging from between 0.10 and 0.16 g/mL of lactic acid during 60 h of fermentation. The above-mentioned values are higher than those obtained in this study.

### 3.3. Lactic Acid Bacteria Count

Results for regards are presented in Figure 2 depicts counts of viable lactic cells. The bacterial population remained stable in fermented plant extracts, only presenting a reduction at the end of the storage period. Most formulations presented viable cell populations above the minimum limits of 10^6^ CFU/mL [8,34] that a probiotic functional food must present during its period of validity, with the exceptions of unadjusted rice and oat extracts and almond and oat extracts in which the soluble solids content was adjusted. These proportions are suggested by researchers as the minimum acceptable amounts at which extracts offer health benefits [8,35]. 

Among the formulations studied, five obtained satisfactory counts for viable lactic cells by the end of the 28-day storage period. The control (milk) presented a high cell count, with a value of 10 log CFU/mL within 28 days. From among the fermentates, soybean, and Brazil nuts, both with and without adjustment in soluble solids, obtained the highest cell counts during the storage period. Soybean remained stable for 21 days; after this period a reduction of 1 log cycle was observed during the 28 days. Lactic acid bacteria populations were higher than those obtained by Mohammadi et al. [29], which obtained values below 7 log CFU/g during 21 days of storage. Battistini et al. [26] recorded populations below 9 log CFU/mL within a 28-day period. The two formulations of Brazil nuts fermentate did not differ from one another until the twenty-first day of storage. After this period, the fermentate for which an adjustment in soluble solids was carried out saw a one logarithm cycle reduction. The adjusted fermentate presented a reduction of two logarithm cycles during the 28-day period. Romano et al. [36] assessed the viability of *Lactobacillus rhamnosus* strains in a Brazil nuts mousse for 3 months at 15 °C and obtained a viability of 10 log CFU/g.

Fermented oat extract under both conditions (with and without adjustment of soluble solids) presented a reduction of two logarithmic cycles at 21 days of storage. At 28 days, there was a decline in the viability of lactic acid bacteria, which subsequently reached unsatisfactory values (Figure 2). Bernat et al. [16] obtained values of 7 log CFU/mL in 7 days of storage and this value was maintained for a period of 28 days.

Fermented rice extract under both conditions (with and without adjustment of soluble solids) presented high viable cell counts at the beginning of the storage period; however, the unadjusted fermentate reduced a logarithm cycle at 21 days and presented a reduction in viable lactic cells at 28 days (Figure 2). Adjusted fermented rice extract presented a reduction of two logarithmic cycles at 21 days of storage, and viable lactic cells were maintained up until the 28th day with 8 log CFU/ mL. Similar values were reported by Jukonyte et al. [33] who studied the fermentation performance of lactic acid bacteria applied to a prepared rice powder at a 1:3 ratio at 30 °C during a 48-h fermentation period, presenting viable cell counts of up to 9.6 log CFU/mL.

The fermented rice extract in which soluble solids were not adjusted did not present viable lactic cell counts at the end of the 28-day storage period (Figure 2). Conversely, the adjusted fermented extract maintained viable cells through the 28-day storage period. Such results may be related to the composition of the substrates and this suggests that sucrose supplementation in the rice extract possibly influenced the development of lactic acid bacteria.

With regards to almond extracts, it was observed that the fermentate in which soluble solids were not adjusted maintained a stable number of viable cells during the 28-day period (Figure 2). The adjusted fermentate experienced sharp decreases during storage with reductions of one logarithm cycle at 7 and 14 days and at 21 days presented values of 1 log CFU/mL. Bernat et al. [27] reported higher values in an almond fermentate supplemented with inulin, with counts ranging from between 6 and 7 log CFU/mL over a 28-day storage period.

### 3.4. Composition of Plant-Based Extracts

The percentage composition of the plant extracts is shown in Table 4. Lipid content did not generally vary with the fermentation process in relation to raw materials. It was observed that the addition of sucrose did not influence the extracts. Significant variations only occur as a result of the different raw materials used.

Bernart et al. [27], when evaluating a supplemented almond extract, reported lipid contents of 3.96%, higher than those shown in Table 4. Bianchi et al. [37] obtained lower values, 1.43% for lipids, compared to the values presented in this study. Rice extracts did not present variations relative to one another. Puerari et al. [32], obtained a value of 0.01% lipids in a rice-based fermented beverage, a value lower than those presented in Table 4. With regards to lipids in oat extracts, a significant difference was not observed among the different conditions studied. Bernat et al. [16], when studying the characterization of an oat extract, discovered a 0.094% lipid quantity. This value is lower than those obtained in this study.

With regards to proteins, it was observed that significant variation only occurred between the different raw materials used, but they were statistically equal when compared to unfermented and fermented substrates. The control sample presented the highest protein content, as expected (Table 4).

Almond and Brazil nuts presented the highest protein indexes among the plant extracts in the study. Values ranging from 1.84% to 1.96% proteins were obtained for almonds. Protein values ranging from 1.13% to 1.40% were obtained for Brazil nuts extract. Bernat et al. [27] analyzed the composition of an almond extract and obtained values that were lower than 1.37% protein. With regard to protein levels in soy extracts, a significant difference was not observed between the substrate and the fermentate (Table 4). Bianchi et al. [37] obtained values that were similar to those found during this study when analyzing a water-soluble soybean extract. 

Oat extracts did not differ from one another with regard to proteins. Protein values ranged from 0.76% to 1.05%. Bernat et al. [16] obtained a lower protein value of 0.65% when characterizing an oat extract. On the other hand, rice extracts presented the lowest observed levels of protein contents and did not show variations between one another (Table 4). Puerari et al. [32] determined the protein content in a rice-based fermented beverage and uncovered similar values, ranging from between 0.39% and 0.42% proteins. 

The main differences in ash content presented were significant depending on the composition of the raw material used (Table 4). The fermentation process significantly altered the ash content in control samples (milk) and soybean extracts, which presented a reduction in ash (Table 4). Almond extracts did not present a significant variation, with the exception of the extract in which soluble solids content was adjusted at 4 °Brix (Almonds WA). Bernat et al. [27] found a value of 0.32% of ashes in almond extract, which is higher than those of the present study. Brazil nuts extracts also did not present variations in the ash parameter relative to one another (Table 4).

Rice extracts did not present variations among one another for the ash parameter. Puerari et al. [32], when characterizing a rice-based fermented beverage, obtained average values ranging between 0.02% and 0.03% ash, values which are similar to those obtained in this study. For oat extracts, a reduction in ash content was observed without a significant difference in the conditions studied (Table 4). Bernat et al. [16] reported a higher value than those obtained in this study.

With regards to dry extracts, no considerable effect was observed after fermentation (Table 4). Observable differences only occurred between the different raw materials used. Additionally, different levels of dry extract were observed as a function of sucrose supplementation. The control sample (milk) had the highest content of dry extract, with values ranging from between 9.87 and 10.91%.

The dry extract in the almond extracts presented a significant difference between the substrate adjusted with sucrose and the substrate that was not subject to adjustment. With regard to oat extracts, dry extract values ranged from 5.96% to 8.80%, with a significant difference observed between extracts with and without sucrose (Table 4). Bernat et al. [16], while studying the development of an oat extract, reported a value of 6.5% dry extract.

The lowest dry extract contents were observed in soybean extracts, which ranged from 3.15% to 3.17% (Table 4). These dry extracts did not present a significant difference. Bianchi et al. [37] reported higher values, with 6.33% dry extract. A significant difference was only observed between Brazil nuts extracts with and without sucrose adjustment, as shown in Table 4. With regards to the dry extract for rice extracts, a significant variation was observed between conditions with and without adjustment of soluble solids. Puerari et al. [32] characterized a rice-based fermented beverage and obtained values for a dry extract that ranged between 8.16% for the unfermented beverage and 8.99% after fermentation was complete.

With regards to carbohydrates, the control sample (milk) presented the highest glucose content (0.38%) and a significant reduction after fermentation. With regards to the remaining extracts, the oat extract obtained the highest glucose contents (Table 4). There are no significant differences in glucose levels observed among the almond extracts. However, a significant variation in fructose and sucrose in adjusted substrates after fermentation was confirmed. Bernat et al. [27], in a study of plant-based almond extract, observed a total sugar content of 0.1285%, which was lower than those obtained in this study.

With regards to Brazil nuts extracts, it can generally be observed that variation between the samples did not occur (Table 4); however, the adjusted fermented sample presented an increase in glucose and fructose content after fermentation, with a significant difference observed between samples of the same substrate. A reduction in fructose was observed in rice extracts after fermentation (Table 4). Additionally, there was no significant difference observed between raw material samples and fermented samples. The differences observed are only associated with samples supplemented with sucrose. Carvalho et al. [17] obtained a total carbohydrate content of 3.04% and 3.16% in water-soluble brown rice and broken rice extracts, respectively.

Soybean extracts obtained the highest levels of fructose and sucrose from among the unadjusted plant extracts studied (Table 4). A significant difference in fructose and sucrose content was not observed after fermentation. Bianchi et al. [37] obtained a total carbohydrate content of 1.68% when characterizing soybean extract.

As expected, reductions in the extracts’ sugar content were generally observed after fermentation. Few reductions in glucose and fructose content and larger reductions in sucrose were observed in adjusted extracts. Such changes may justify the development and maintenance of lactic acid bacteria in beverages during the fermentation process.

## 4. Conclusions

Plant extracts in which the soluble solid content was adjusted at 4.0 °Brix largely presented significant variation in their physicochemical characteristics after fermentation when compared to extracts that were not adjusted. Physicochemical characterization and analysis of the percentage composition produced results that were similar to the existing literature. Variations were observed as a function of the extracts’ formulation, the raw materials used, and the manner in which the water-soluble plant extracts were obtained. The viability of lactic acid bacteria was satisfactory for fermented plant-based almond, Brazil nuts, and soybean extracts in which soluble solid content was not adjusted and rice and Brazil nuts in which soluble solids were adjusted to 4.0 °Brix. The adjustment of soluble solids to 4.0 °Brix did not influence the rate of development and stability of lactic acid bacteria. This study therefore demonstrated that completely replacing milk with water-soluble plant extracts offers great potential in the development of functional fermented beverages. This study demonstrated that certain fermented plant-based extracts have high potential in the development of new products that are analogous to traditional dairy products for meeting a variety of dietary needs and personal preferences. The fermentation of plant-based products also contributes to diversification and innovation in the food industry.

## Figures and Tables

**Figure 1 foods-12-04128-f001:**
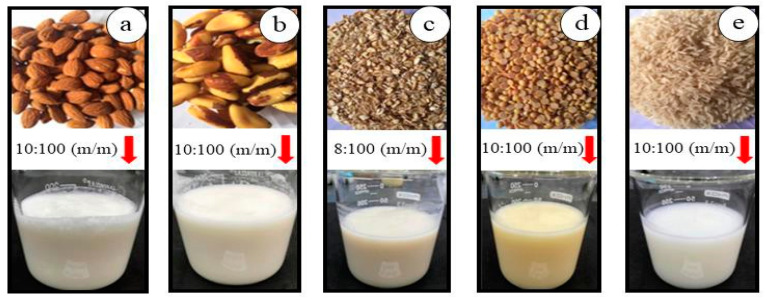
(**a**) Almond extract, (**b**) Brazil nuts extract, (**c**) oat extract, (**d**) soybean extract, and (**e**) rice extract.

**Figure 2 foods-12-04128-f002:**
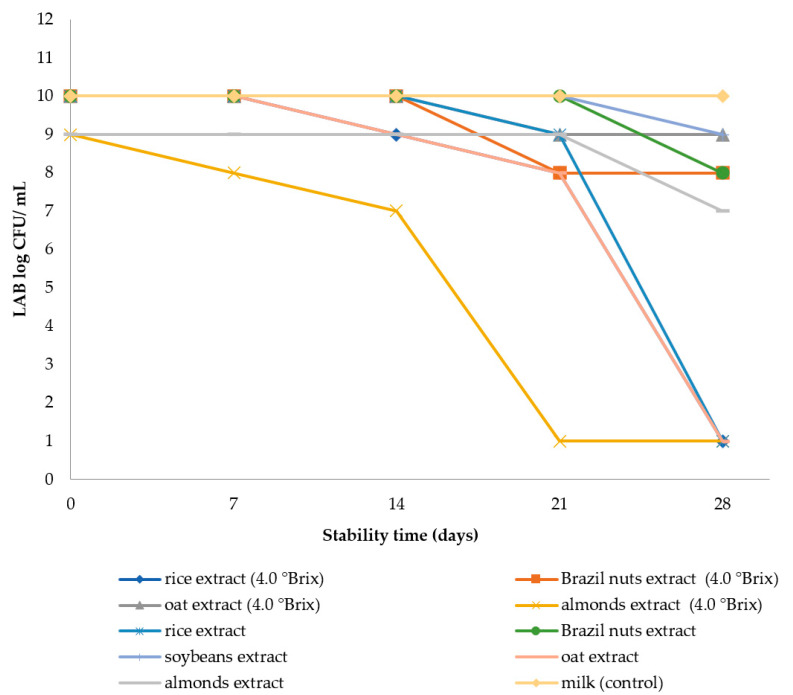
Lactic acid bacteria (LAB) counts in fermented plant extracts with and without soluble solids adjustment over 28 days of stability.

**Table 1 foods-12-04128-t001:** Values of pH, titratable acidity, and soluble solids content are determined at the beginning and end of fermentation.

Parameters	Sample	Initial (0 h)	Final (12 h)
pH	Milk (control)	6.60 ± 0.02 ^c^	4.26 ± 0.01 ^f^
	Almonds	6.61 ± 0.07 ^c^	4.36 ± 0.01 ^e^
	Rice	6.47 ± 0.09 ^d^	4.92 ± 0.06 ^b^
	Oats	6.65 ± 0.05 ^c^	4.97 ± 0.05 ^a^
	Brazil nuts	7.09 ± 0.05 ^a^	4.71 ± 0.04 ^c^
	Soybean	6.76 ± 0.06 ^b^	4.41 ± 0.02 ^d^
	Almond WA	6.45 ± 0.03 ^d^	4.24 ± 0.03 ^f,g^
	Rice WA	6.45 ± 0.04 ^d^	4.15 ± 0.02 ^h^
	Oats WA	6.50 ± 0.15 ^d^	4.10 ± 0.01 ^h^
	Brazil nuts WA	7.07 ± 0.03 ^a^	4.23 ± 0.02 ^g^
Titratable acidity (% lactic acid)	Milk (control)	0.17 ± 0.01 ^a^	0.87 ± 0.01 ^a^
	Almond	0.01 ± 0.00 ^e^	0.33 ± 0.01 ^b^
	Rice	0.02 ± 0.01 ^d,e^	0.02 ± 0.00 ^g^
	Oats	0.04 ± 0.00 ^b,c,d^	0.15 ± 0.01 ^e^
	Brazil nuts	0.06 ± 0.00 ^b^	0.30 ± 0.00 ^c^
	Soybean	0.05 ± 0.01 ^b,c^	0.27 ± 0.09 ^d^
	Almond WA	0.03 ± 0.00 ^c,d,e^	0.33 ± 0.01 ^b^
	Rice WA	0.02 ± 0.01 ^d,e^	0.05 ± 0.01 ^f^
	Oats WA	0.04 ± 0.01 ^b,c,d^	0.16 ± 0.01 ^e^
	Brazil nuts WA	0.03 ± 0.00 ^c,d,e^	0.32 ± 0.01 ^b,c^
Soluble solids content (°Brix)	Milk (control)	13.00 ± 0.00 ^a^	7.00 ± 0.01 ^a^
	Almond	1.00 ± 0.00 ^c^	1.00 ± 0.00 ^e^
	Rice	1.00 ± 0.00 ^c^	1.00 ± 0.00 ^e^
	Oats	1.00 ± 0.00 ^c^	1.00 ± 0.01 ^e^
	Brazil nuts	1.00 ± 0.00 ^c^	1.00 ± 0.00 ^e^
	Soybean	4.00 ± 0.00 ^b^	1.00 ± 0.00 ^e^
	Almond WA	4.00 ± 0.00 ^b^	2.00 ± 0.00 ^d^
	Rice WA	4.00 ± 0.00 ^b^	4.00 ± 0.00 ^b^
	Oats WA	4.00 ± 0.00 ^b^	4.00 ± 0.00 ^b^
	Brazil nuts WA	4.00 ± 0.00 ^b^	3.00 ± 0.01 ^c^

The means of 3 replicates followed by the same letter, in the same column, are not different (*p* > 0.05). WA = With the adjustment in soluble solids at 4.0 °Brix.

**Table 2 foods-12-04128-t002:** pH values of fermented plant extracts stored at 5 °C for 28 days.

Sample			Time (Days)		
	0	7	14	21	28
Milk (control)	4.37 ± 0.06 ^b^	4.38 ± 0.05 ^c^	4.35 ± 0.08 ^c^	4.53 ± 0.03 ^a^	4.15 ± 0.05 ^d^
Almond	4.05 ± 0.03 ^d^	4.03 ± 0.02 ^g^	4.12 ± 0.01 ^e^	4.14 ± 0.02 ^d^	4.15 ± 0.01 ^d^
Rice	4.07 ± 0.04 ^d^	4.08 ± 0.02 ^f^	4.42 ± 0.08 ^b,c^	4.29 ± 0.24 ^c^	5.08 ± 0.14 ^a^
Oats	4.17 ± 0.07 ^c^	4.13 ± 0.06 ^d,e^	4.16 ± 0.08 ^d,e^	4.27 ± 0.06 ^c^	4.38 ± 0.08 ^c^
Brazil nuts	4.36 ± 0.02 ^b^	4.44 ± 0.03 ^b^	4.44 ± 0.03 ^b^	4.45 ± 0.03 ^a,b^	4.43 ± 0.03 ^c^
Soybean	4.50 ± 0.02 ^a^	4.64 ± 0.02 ^a^	4.57 ± 0.03 ^a^	4.41 ± 0.01 ^b^	4.66 ± 0.04 ^b^
Almond WA	4.06 ± 0.02 ^d^	4.06 ± 0.02 ^d^	4.19 ± 0.01 ^d,e^	4.21 ± 0.01 ^c,d^	4.18 ± 0.03 ^d^
Rice WA	3.68 ± 0.12 ^f^	3.68 ± 0.05 ^i^	3.94 ± 0.21 ^f^	3.40 ± 0.00 ^f^	4.00 ± 0.10 ^e^
Oats WA	3.82 ± 0.07 ^e^	3.88 ± 0.06 ^h^	3.83 ± 0.07 ^g^	3.89 ± 0.03 ^e^	3.71 ± 0.09 ^f^
Brazil nuts WA	4.13 ± 0.02 ^c^	4.16 ± 0.01 ^d^	4.22 ± 0.02 ^d^	4.24 ± 0.02 ^c^	4.22 ± 0.02 ^d^

The means of 3 replicates followed by the same letter, in the same column, are not different (*p* > 0.05). WA = With the adjustment in soluble solids at 4.0 °Brix.

**Table 3 foods-12-04128-t003:** Titratable acidity (% lactic acid) of fermented plant extracts during 28-day storage at 5 °C.

Sample			Time (Days)		
	0	7	14	21	28
Milk (control)	0.93 ± 0.05 ^a^	0.86 ± 0.01 ^a^	0.85 ± 0.01 ^a^	0.83 ± 0.01 ^a^	0.80 ± 0.01 ^a^
Almond	0.40 ± 0.00 ^c^	0.40 ± 0.02 ^d^	0.43 ± 0.01 ^c^	0.44 ± 0.04 ^d^	0.42 ± 0.02 ^d^
Rice	0.03 ± 0.01 ^h^	0.03 ± 0.01 ^i^	0.03 ± 0.00 ^g^	0.02 ± 0.00 ^g^	0.02 ± 0.00 ^h^
Oats	0.09 ± 0.01 ^f^	0.10 ± 0.01 ^g^	0.10 ± 0.00 ^f^	0.11 ± 0.01 ^f^	0.10 ± 0.01 ^g^
Brazil nuts	0.20 ± 0.00 ^e^	0.25 ± 0.01 ^f^	0.27 ± 0.01 ^e^	0.35 ± 0.03 ^e^	0.32 ± 0.04 ^f^
Soybean	0.36 ± 0.00 ^d^	0.37 ± 0.01 ^e^	0.34 ± 0.00 ^d^	0.34 ± 0.01 ^e^	0.35 ± 0.01 ^e^
Almond WA	0.48 ± 0.03 ^b^	0.52 ± 0.04 ^b^	0.59 ± 0.05 ^b^	0.50 ± 0.02 ^b^	0.53 ± 0.01 ^b^
Rice WA	0.05 ± 0.01 ^g^	0.02 ± 0.00 ^i^	0.02 ± 0.00 ^b^	0.02 ± 0.00 ^g^	0.02 ± 0.00 ^h^
Oats WA	0.10 ± 0.00 ^f^	0.08 ± 0.01 ^h^	0.09 ± 0.01 ^f^	0.11 ± 0.01 ^f^	0.10 ± 0.01 ^g^
Brazil nuts WA	0.41 ± 0.02 ^c^	0.43 ± 0.01 ^c^	0.44 ± 0.02 ^c^	0.48 ± 0.01 ^c^	0.48 ± 0.01 ^c^

Means followed by the same letter in the columns are not different (*p* > 0.05) by the Tukey test. WA = With the adjustment in soluble solids at 4.0 °Brix.

**Table 4 foods-12-04128-t004:** Percentage composition of plant extracts before and after a 12 h fermentation.

Samples	Dry Extract (%)	Lipids (%)	Ash (%)	Proteins (%)		Carbohydrates	
					Glucose (%)	Fructose (%)	Sucrose (%)
Milk	10.91 ± 0.25 ^a^	2.78 ± 0.22 ^a^	0.88 ± 0.02 ^a^	3.44 ± 0.04 ^a^	0.38 ± 0.06 ^a^	0.06 ± 0.10 ^h,i^	0.04 ± 0.06 ^g^
Milk F	9.87 ± 0.20 ^b^	2.13 ± 0.08 ^a,b^	0.78 ± 0.03 ^b^	3.32 ± 0.08 ^a^	0.18 ± 0.04 ^c,d^	0.04 ± 0.00 ^i^	0.02 ± 0.01 ^g^
Almond	6.53 ± 0.18 ^e^	2.74 ± 0.22 ^a^	0.09 ± 0.01 ^e^	1.96 ± 0.12 ^b^	0.07 ± 0.04 ^d,e^	0.30 ± 0.52 ^c,d^	0.004 ± 0.01 ^g^
Almond F	6.52 ± 0.08 ^e^	2.63 ± 0.10 ^a,b^	0.09 ± 0.02 ^e^	1.84 ± 0.11 ^b^	0.10 ± 0.06 ^c,d,e^	0.38 ± 0.12 ^b,c^	-
Almond WA	10.69 ± 0.17 ^a^	2.49 ± 0.09 ^a,b^	0.14 ± 0.04 ^d^	1.93 ± 0.01 ^b^	0.18 ± 0.59 ^c,d^	0.49 ± 0.11 ^a^	3.33 ± 0.31 ^a^
Almonds WAF	10.35 ± 0.52 ^a,b^	2.50 ± 0.54 ^a,b^	0.11 ± 0.01 ^d,e^	1.87 ± 0.11 ^b^	0.10 ± 0.04 ^c,d,e^	0.37 ± 0.07 ^b,c^	0.98 ± 0.61 ^e^
Rice	4.31 ± 0.07 ^g^	0.05 ± 0.03 ^d^	0.02 ± 0.00 ^f^	0.4 ± 0.00 ^h^	0.13 ± 0.01 ^c,d,e^	0.01 ± 0.01 ^i^	-
Rice F	4.28 ± 0.03 ^g^	0.05 ± 0.01 ^d^	0.02 ± 0.01 ^f^	0.45 ± 0.10 ^h^	0.01 ± 0.01 ^e,f^	0.01 ± 0.00 ^i^	-
Rice WA	7.50 ± 0.16 ^d^	0.04 ± 0.02 ^d^	0.03 ± 0.01 ^f^	0.42 ± 0.00 ^h^	0.15 ± 0.33 ^c,d^	0.11 ± 0.00 ^g,h,i^	2.76 ± 1.02 ^c^
Rice WAF	7.51 ± 0.05 ^d^	0.06 ± 0.02 ^d^	0.02 ± 0.01 ^f^	0.43 ± 0.08 ^h^	0.16 ± 0.22 ^c,d^	0.10 ± 0.01 ^g,h,i^	3.07 ± 0.93 ^b^
Oats	5.97 ± 0.13 ^e,f^	0.38 ± 0.06 ^c,d^	0.01 ± 0.01 ^f^	0.97 ± 0.03 ^e,f,g^	0.17 ± 0.42 ^c,d^	0.34 ± 0.51 ^b,c,d^	-
Oats F	6.15 ± 0.12 ^e,f^	0.26 ± 0.03 ^c,d^	0.01 ± 0.00 ^f^	1.05 ± 0.07 ^c,d,e,f,g^	0.17 ± 0.20 ^c,d^	0.26 ± 0.06 ^d,e,f^	-
Oats WA	8.80 ± 0.31 ^c^	0.34 ± 0.02 ^c,d^	0.01 ± 0.00 ^f^	0.76 ± 0.01 ^g,h^	0.35 ± 0.58 ^a,b^	0.18 ± 0.28 ^e,f,g^	2.91 ± 0.07 ^b,c^
Oats WAF	8.31 ± 0.16 ^c^	0.33 ± 0.01 ^c,d^	0.01 ± 0.00 ^f^	0.76 ± 0.12 ^g,h^	0.23 ± 0.33 ^b,c^	0.17 ± 0.04 ^f,g^	2.79 ± 0.25 ^c^
** Brazil nut	5.04 f ± 0.11 ^f,g^	2.43 ± 0.09 ^a,b^	0.10 ± 0.01 ^f^	1.32 ± 0.03 ^c,d,e^	0.01 ± 0.01 ^e^	0.26 ± 0.01 ^d,e,f^	-
** Brazil nut F	5.39 ± 0.20 ^f^	2.07 ± 0.20 ^a,b^	0.07 ± 0.01 ^e^	1.13 ± 0.01 ^c,d,e,f^	0.01 ± 0.01 ^e^	0.24 ± 0.04 ^d,e,f^	-
** Brazil nut WA	8.74 ± 0.08 ^c^	2.30 ± 0.04 ^a,b^	0.13 ± 0.01 ^d,e^	1.38 ± 0.02 ^c,d^	0.01 ± 0.01 ^e^	0.26 ± 0.13 ^d,e,f^	2.96 ± 0.04 ^b,c^
** Brazil nut WAF	8.57 ± 0.27 ^c^	2.54 ± 0.09 ^a,b^	0.12 ± 0.01 ^d,e^	1.40 ± 0.01 ^c^	0.23 ± 0.12 ^b,c^	0.33 ± 0.05 ^b,c,d^	1.92 ± 0.50 ^d^
Soybean	3.15 ± 0.23 ^g^	1.80 ± 0.07 ^b,c^	0.19 ± 0.01 ^c^	0.89 ± 0.01 ^f,g^	0.07 ± 0.11 ^d,e^	0.40 ± 0.03 ^a,b^	0.31 ± 0.11 ^f^
Soybean F	3.17 ± 0.14 ^g^	1.14 ± 0.02 ^c^	0.10 ± 0.00 ^d,e^	0.98 ± 0.02 ^d,e,f,g^	-	0.14 ± 0.18 ^g,h^	-

Means of 3 replicates followed by the same letter, in the same column, are not different *p* ≥ 0.05. (-) undetectable concentration content below 1 mg/L, ** = Brazil nut, F = No adjustment in fermented soluble solids, WA = With adjustment of soluble solids at 4 °Brix, WAF = With adjustment of soluble solids at 4 °Brix fermented.

## Data Availability

The datasets generated during and/or analyzed during the current study are available from the corresponding author on reasonable request.

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
