# Peer review of "Plant-Based Fermented Beverages: Development and Characterization"

_foods, 2023, doi:10.3390/foods12224128_

Round 1
Reviewer 1 Report
The English of the manuscript requires significant improvement as some sentences are difficult to understand.
The title is misleading. Any beverages to be used as an alternative to milk could have meant that the beverages could substitute/replace milk. It is generally known that none of the plant-based beverages except soybean could have comparable nutritional composition similar to milk. Please remove 'milk' from the title and rephrase the title to reflect the content of the manuscript.
The fermentable ability as a substrate is not a criterion to be considered as an alternative to cow milk. Please correct the facts in the introduction and discussion section.
Why the commercial starter used is named as 'milk yeast'? What yeast strain is contained in the starter?
In the preparation of plant solution/beverage, what is the ratio of cereal: water or legume: water? The ratios including the added water during cooking must be clearly described for every substrate.
The concentration of starter culture used at 400 mg/L beverage is to achieve a standardize inoculum. What is the initial count (CFU/mL) to be achieved prior to the fermentation?
Table 1 - There is no point to compare the values of the different samples prior to the fermentation especially if the values are too small (e.g. acidity). In fact, the comparison should be in between before (0h) and after fermentation (12h).
Table 2 - The significant differences should be between the interval storage time (within the same row), not the same column. It is meaningless to compare the same column.
The results and discussions for the above Tables (1-2) should be rewritten due to the changes in statistical analysis of the measurements.
Fig. 2 - Use same color for the same extracts (with or without adjustment), but differentiated by different symbols. This could lead to a better observation by the readers.
From the body of knowledge, please explain why the LAB counts for the extracts without added sucrose reduced more than 7 log CFU within 3-4 weeks of storage after fermentation. The sugar could have been used during fermentation.
Comparing composition of unfermented and fermented products (Table 4) is meaningless as the results are expected. The rationale of having Table 4 must be justified.
Please explain why the % sucrose of the beverages with added sucrose was found significantly lower than the control? This seems contradict to the LAB count during storage especially for beverages with adjusted sucrose?
The conclusion should be rewritten to highlight the new information found from the present study. What is the usefulness of the findings to the industry? It has nothing to do with replacing milk.
The manuscript requires extensive language editing.
Author Response
REVIEWER #1:
- The English of the manuscript requires significant improvement as some sentences are difficult to understand.
Answer: The article has been throughout revised both by the authors and a native English speaker.
- The title is misleading. Any beverages to be used as an alternative to milk could have meant that the beverages could substitute/replace milk. It is generally known that none of the plant-based beverages except soybean could have comparable nutritional composition similar to milk. Please remove 'milk' from the title and rephrase the title to reflect the content of the manuscript.
Answer: The itle has been modified. “Plant-based fermented beverages: development and characterization”.
- The fermentable ability as a substrate is not a criterion to be considered as an alternative to cow milk. Please correct the facts in the introduction and discussion section.
Answer: Indeed, we were mistaken in using the word substitute, the word alternative is more appropriate. The fact that plant-based milks behaved well in the fermentation process, using lactic acid bacteria (LAB), does not qualify them as substitutes for milk of animal origin or derivatives. But it expands the possibilities of applying LAB, on bases other than dairy bases. Since plant-based milks are used as alternatives to milk of animal origin, on several occasions, especially by individuals who are allergic to milk, lactose intolerant or have vegetarian and vegan habits or just due to dietary preferences. In this way, plant-based milk can be used in a similar way to cow's milk, in recipes, desserts, sauces and other applications. And fermented vegetable milks are another technological application within the universe of plant-based products. In this way, the corrections were accepted.
- Why the commercial starter used is named as 'milk yeast'? What yeast strain is contained in the starter?
Answer: The correction has been made. The error occurred in the translation. No strain of yeast was used, the starter used contained only strains of lactic acid bacteria, such as Streptococcus thermophilus, Lactobacillus acidophilus LA-5® and Bifidobacterium BB-12.
- In the preparation of plant solution/beverage, what is the ratio of cereal: water or legume: water? The ratios including the added water during cooking must be clearly described for every substrate.
Answer: As requested by the reviewers, the material and methods section was revised and adjustments were made, so that there is better understanding for the reader.
- The concentration of starter culture used at 400 mg/L beverage is to achieve a standardize inoculum. What is the initial count (CFU/mL) to be achieved prior to the fermentation?
Answer: In response to the request, the initial concentrations of the starters were added to the article. Thus providing complete information, thus allowing the reproducibility of the methods.
- Table 1 - There is no point to compare the values of the different samples prior to the fermentation especially if the values are too small (e.g. acidity). In fact, the comparison should be in between before (0h) and after fermentation (12h).
- Table 2 - The significant differences should be between the interval storage time (within the same row), not the same column. It is meaningless to compare the same column.
- The results and discussions for the above Tables (1-2) should be rewritten due to the changes in statistical analysis of the measurements.
Answer to questions 7, 8 and 9: Dear reviewer, we are very grateful for the comments and suggestions made to improve the quality of this study. Regarding the statistical analysis of the data, in the present study a Completely Randomized Design (DIC) was adopted, in which the data were subjected to analysis of variance (ANOVA), being evaluated by the Tukey mean test with 95% confidence (p < 0.05). In order to understand the behavior of vegetable drinks in refrigerated storage, the statistical treatment was carried out by applying the comparison of means between treatments, that is, which treatments are different or the same based on the control formulation during refrigerated storage for 28 days. On the contrary, the focus is not on observing the particular or specific behavior of each treatment during storage. In view of this, we authors understand that statistical analysis between the different formulations on refrigerated storage days is fully appropriate, with the comparison of means in the columns remaining. So, again, we appreciate the suggestions given by the reviewer.
- Fig. 2 - Use same color for the same extracts (with or without adjustment), but differentiated by different symbols. This could lead to a better observation by the readers.
Answer: We appreciate the suggestion, but we understand that the use of different colors to represent different samples in a graph has the objective of visibly distinguishing the samples and facilitating the understanding of the data and helping the reader to make a faster connection between the information.
- From the body of knowledge, please explain why the LAB counts for the extracts without added sucrose reduced more than 7 log CFU within 3-4 weeks of storage after fermentation. The sugar could have been used during fermentation.
Answer: LAB counts can reduce in fermented products over storage time, this so-called decay phase is a natural process of a microbial population. And the conditions that lead to this phase may be a lack of substrates, as the bacteria continue to grow and multiply, the substrates available in the environment are consumed. When essential substrates become scarce, microorganisms no longer have the nutrients necessary for their development. Another factor is the accumulation of metabolites, during the exponential phase, bacteria can produce different metabolites, in the case of LAB the production of lactic acid, which can become toxic to the microorganisms themselves. The accumulation of these products in the prolonged environment can inhibit cell development. Likewise, changes in pH, temperature and other factors can make the environment unsuitable for the development and maintenance of cells for a long period.
The study evaluated the fermentation process and the stability of drinks obtained with and without added sugar.
- Comparing composition of unfermented and fermented products (Table 4) is meaningless as the results are expected. The rationale of having Table 4 must be justified.
Answer: We acknowledge the reviewer's concern. However, presenting the composition of vegetable milk before and after fermentation is necessary, since in order to characterize a new fermented drink, it is necessary to present possible changes in its nutritional profile. Therefore, we understand that table 4 is relevant and its presentation is important.
- Please explain why the % sucrose of the beverages with added sucrose was found significantly lower than the control? This seems contradict to the LAB count during storage especially for beverages with adjusted sucrose?
Answer: In this study, the control drink is milk of animal origin and when comparing fermented milk after 12 hours, it did not present higher sucrose values than the fermented vegetable milks analyzed. For example: after 12 hours of fermentation, fermented animal milk obtained 0.02% sucrose, while the vegetable milk that presented the lowest percentage of sucrose after 12 hours of fermentation was soy milk with 0.31% followed by vegetable milk. of almonds with 0.98% sucrose. Therefore, the counts are as expected.
- The conclusion should be rewritten to highlight the new information found from the present study. What is the usefulness of the findings to the industry? It has nothing to do with replacing milk.
Answer: We appreciate your suggestions, but there were no major changes in our results. The conclusion was updated due to the suggestion given, how this research would be useful for the industry and the suitability of the term milk replacement. Corrections were made to the text.
Reviewer, thank you very much for your contribution to this study!
Reviewer 2 Report
I reviewed the manuscript titled “Functional Plant-Based Beverage: An Alternative to Milk”. The manuscript is well written and contributes to the field. In my opinion, this version needs a couple of revisions before being accepted for possible publication consideration.
Title can be revised. As such, I had an impression of the review article. Please revise the title according to the content described in the article.
Line 19: did not… please use more appropriate word
Keywords: add new word such as plant-based milk
Introduction
Authors also characterize the product. If so, this should be included in objectives
Materials and methods
All these methods are clearly described
Results and discussion
3.1. Plant extract fermentation
Table 1. a, b, c, must be in superscript after the standard deviation
The discussion in section 3.1. should be improved
Table 2. comparison with time (days) should be provided with statistical comparison
3.2. Physicochemical stability: discussion is appropriate
Table 3. a, b, c, must be in superscript after the standard deviation and comparison with time (days) should be provided with statistical comparison
Line 321: Results for regards are presented in Figure 2 depicts counts of viable lactic cells. The English language is very poor. Sentence is wrong
Figure 2: quality must be improved. For example, X axis t(d). what it is, time (days)?. Kindly write it appropriately. remove the border line
3.4. Composition of plant extracts. Is it Plant-based milk?
Minor editing is required. Some sentences must be revised
Author Response
REVIEWER #2:
- I reviewed the manuscript titled “Functional Plant-Based Beverage: An Alternative to Milk”. The manuscript is well written and contributes to the field. In my opinion, this version needs a couple of revisions before being accepted for possible publication consideration.
Answer: We thank you for the notes and suggestions that made this article better.
- Title can be revised. As such, I had an impression of the review article. Please revise the title according to the content described in the article.
Answer: We authors agreed to change the title of the article, making it more understandable. “Plant-based fermented beverages: development and characterization”.
- Line 19: did not… please use more appropriate word.
Answer: We consider the correction valid. The correction was made to the text.
- Keywords: add new word such as plant-based milk.
Answer: We add the suggested keyword based on the reviewer's requested consideration.
- Authors also characterize the product. If so, this should be included in objectives.
Answer: Thank you very much for the suggestion, the information was included in the text.
- Table 1. a, b, c, must be in superscript after the standard deviation.
Answer: We recognized the error, the note was accepted and corrections were made to the table.
- The discussion in section 3.1. should be improved.
Answer: Thank you for your note. In general, we understand that the points discussed are in agreement. Please point out where we can improve.
- Table 2. comparison with time (days) should be provided with statistical comparison.
- Table 3. a, b, c, must be in superscript after the standard deviation and comparison with time (days) should be provided with statistical comparison.
Answer to questions 8 and 9: The reviewer's request regarding the adequacy of the subscript letters was made. Regarding the statistical analysis of the data, in the present study a Completely Randomized Design (DIC) was adopted, in which the data were subjected to analysis of variance (ANOVA), being evaluated by the Tukey mean test with 95% confidence (p < 0.05). In order to understand the behavior of vegetable drinks in refrigerated storage, the statistical treatment was carried out by applying the comparison of means between treatments, that is, which treatments are different or the same based on the control formulation during refrigerated storage for 28 days. On the contrary, the focus is not on observing the particular or specific behavior of each treatment during storage. In view of this, we authors understand that statistical analysis between the different formulations on refrigerated storage days is fully appropriate, with the comparison of means in the columns remaining. So, again, we appreciate the suggestions given by the reviewer.
- Line 321: Results for regards are presented in Figure 2 depicts counts of viable lactic cells. The English language is very poor. Sentence is wrong.
Answer: The article was subjected to a general writing review to eliminate language errors.
- Figure 2: quality must be improved. For example, X axis t(d). what it is, time (days)? Kindly write it appropriately. Remove the border line.
Answer: Figure 2 was modified according to the reviewer's suggestions.
- 3.4. Composition of plant extracts. Is it Plant-based milk?
Answer: The subtitle change was made, we recognize that the term Plant-based milk is more appropriate.
Reviewer, thank you very much for your contribution to this study!
Reviewer 3 Report
Characterization of some plant-based beverages was presented in this study. Authors must state why they chose so many variants. as there are already some similar products on the market. Characterization without sensory analyzing is not a complete one. Please add it as it is the most important for the consumer. The journal Foods is an Open Access one and will be accessed by many industrials interested in developing novel products. They do not understand much from such a presentation of the results. You have to point out better the importance of your findings and the novelty degree.
Few spelling errors were encountered. Please check spelling and grammar carefully.
Author Response
REVIEWER #3:
- Characterization of some plant-based beverages was presented in this study. Authors must state why they chose so many variants. As there are already some similar products on the market.
Answer: During the literature review, a variety of plant-based milks were observed being applied in fermentation processes, with different strains. Therefore, this study aimed to compare different plant sources and evaluate the stability of the fermented drinks obtained using lactic acid bacteria (LAB) in the fermentation process. By including a variety of variants, it allowed to increase the generalizability of the results. This means that research results may be more applicable to a wider range of real-world situations. Yes, there are currently different types of plant-based milks and plant-based drinks. However, there are no options for these matrices, especially Brazil nuts and rice, in the development of beverages fermented by lactic acid bacteria.
- Characterization without sensory analyzing is not a complete one. Please add it as it is the most important for the consumer.
Answer: Thank you very much for the suggestion. This study is under development and the authors chose to release the results soon.
- The journal Foods is an Open Access one and will be accessed by many industrials interested in developing novel products. They do not understand much from such a presentation of the results. You have to point out better the importance of your findings and the novelty degree.
Answer: Thank you for your observation. We understand that presenting new product results and expressing key findings to different audiences is a critical part of the process. But the use of graphs and tables, as presented in this study, makes information more accessible and understandable and can help convey complex data more clearly and concisely.
Regarding the degree of novelty, the development of plant-based drinks and foods is continually growing and aims to meet different demands. The novelty is the use and comparison of ingredients in addition to the traditional soy milk, almond milk and oat milk. In this article we are suggesting Brazil nut milk and rice milk as possible sources in the manufacture of fermented drinks, using strains of lactic acid bacteria. In addition to serving the regional market, offering a product inspired by local cuisine, such as Brazil nuts. Another relevant point of this study is to evaluate and compare the development of sugar-free or low-sugar fermented drinks, thus meeting yet another need for health-conscious consumers.
Reviewer, thank you very much for your contribution to this study!